# Want to train KANS at scale? Now UKAN!

## Abstract

Kolmogorov–Arnold Networks (KANs) have recently emerged as a powerful alternative to traditional multilayer perceptrons. However, their reliance on predefined, bounded grids restricts their ability to approximate functions on unbounded domains. To address this, we present Unbounded Kolmogorov–Arnold Networks (UKANs), a method that removes the need for bounded grids in traditional Kolmogorov–Arnold Networks (KANs). The key innovation of this method is a coefficient-generator (CG) model that produces, on the fly, only the B-spline coefficients required locally on an unbounded symmetric grid. UKANs couple multilayer perceptrons with KANs by feeding the positional encoding of grid groups into the CG model, enabling function approximation on unbounded domains without requiring data normalization. To reduce the computational cost of both UKANs and KANs, we introduce a GPU-accelerated library that lowers B-spline evaluation complexity by a factor proportional to the grid size, enabling large-scale learning by leveraging efficient memory management, in line with recent software advances such as FlashAttention and FlashFFTConv. Performance benchmarking confirms the superior memory and computational efficiency of our accelerated KAN (warpKAN), and UKANs, showing a $3 - 30\times$ speedup and up to $1000\times$ memory reduction compared to vanilla KANs. Experiments on regression, classification, and generative tasks demonstrate the effectiveness of UKANs to match or surpass KAN accuracy. Finally, we use both accelerated KAN and UKAN in a molecular property prediction task, establishing the feasibility of large-scale end-to-end training with our optimized implementation.

## 1 Background

Neural networks (MLPs) are the workhorse of the current AI and Deep Learning revolution, driving advances in computer vision, language models, computational science, and more recently biology and molecular science LeCun et al. (2015); Goh et al. (2017); Schütt et al. (2018); Pandey et al. (2022); Raissi et al. (2017). The universal approximation theorem guarantees that MLPs with enough parameters can fit any function. The widespread adoption of MLPs across various disciplines has led to the emergence of exciting applications such as ChatGPT in the large language model (LLM) domain, and AlphaFold in protein structure prediction Vaswani et al. (2017); Jumper et al. (2021). However, MLPs suffer from a few drawbacks, particularly generalization for regression tasks.

Recently, the Kolmogorov-Arnold networks (KANs) Liu et al. (2024b;a) have gained attention as a promising alternative to traditional MLPs, especially in scientific applications, with novel variants currently under development Bozorgasl & Chen (2024); Genet & Inzirillo (2024); Abueidda et al. (2024); Kiamari et al. (2024); Somvanshi et al. (2024). The KAN architecture is partially based on the Kolmogorov-Arnold representation theorem Kolmogorov (1961); Braun & Griebel (2009), which states that any multivariate function on a bounded domain can be obtained by a finite composition of continuous univariate functions and summation. Mathematically, this can be represented as:

$$f(x) = \sum_{q=1}^{2n+1} \phi_q \left( \sum_{p=1}^{n} \phi_{q,p}(x_p) \right) \tag{1}$$

where $\phi_{p,q} : [0,1] \to \mathbb{R}$ and $\phi_q : \mathbb{R} \to \mathbb{R}$. Eq. 1 implies that we can approximate any function by summation of univariate functions. A B-spline curve is one of the best methods to parameterize any univariate function by learning the coefficients of the B-spline basis function.

To extend the KAN architecture beyond what is described by Eq. 1, Liu et al. (2024b) proposes a new way to build the KAN computational graph with $L$ layers. They assume Eq. 1 is a 2-layer KAN with $n$, $2n$, and 1 nodes. The generalization for $L$ layers starts by defining $n_l$ nodes $\forall l = 0, 1, ..., L$, where $n_l$ is the number of nodes in the $l^{th}$ layer of the computational graph.

The activation function between node $i$ in layer $l$, $(l,i)$ and node $j$ in layer $l+1$, $(l+1, j)$ is denoted by $\phi_{l,j,i}$; the activation value of node $(l+1, j)$ is obtained by summing all incoming post-activation values $\phi_{l,j,i}(x_{l,i})$,

$$x_{l+1,j} = \sum_{i=1}^{n_l} \phi_{l,j,i}(x_{l,i}) \tag{2}$$

In total, there are $n_l n_{l+1}$ activation values and connections between layer $l$ and layer $l+1$. For an input $\mathbf{x} \in \mathbb{R}^{n_0}$, a general KAN network can be written as a composition of $L$ layers,

$$\text{KAN}(\mathbf{x}) = \left(\Phi_{L-1} \circ \Phi_{L-2} \circ \cdots \circ \Phi_1 \circ \Phi_0\right) \mathbf{x}. \tag{3}$$

where $\Phi_l$ is the matrix function of shape $n_{l+1} \times n_l$ with element $(j, i)$ corresponding to $\phi_{l,j,i}$ activation function. In practice, $\phi(x)$ is the sum of the basis function $b(x)$ (MLP with activation function) and the spline function,

$$\phi(x) = w_b b(x) + w_s \text{spline}(x) \tag{4}$$

where $\text{spline}(x)$ is parameterized as a linear combination of B-splines such that

$$\text{spline}(x) = \sum_i c_i B_i(x) \tag{5}$$

where $c_i$ are learned parameters. A B-spline of order $k+1$ is a collection of piecewise polynomial functions $B_{i,k+1}(t)$ of degree $k$. The locations where these piecewise polynomials connect to each other are known as knots. Given $m+1$ knot values with a uniqueness constraint on $B_{i,k+1}$, we have,

$$B_{i,k+1}(t) = \begin{cases} \text{non-zero}, & \text{if } t_i \le t < t_{i+k+1} \\ 0, & \text{otherwise} \end{cases} \tag{6}$$

The Cox–de Boor formula recursively builds B-spline of order $k$ using,

$$B_{i,k}(t) = \frac{t - t_i}{t_{i+k} - t_i} B_{i,k-1}(t) + \frac{t_{i+k+1} - t}{t_{i+k+1} - t_{i+1}} B_{i+1,k-1}(t), \tag{7}$$

where $B_{i,0}(t) = \mathbf{1}_{[t_i, t_{i+1}]}(t)$.

KANs are promising, but practical use is hampered by compute- and memory-inefficient implementations and reliance on periodic grid updates. In naïve GPU code, evaluating B-spline bases via the Cox–de Boor recursion redundantly recomputes along the entire knot vector (effectively repeating work $k+1$ times per evaluation) and inflates both memory traffic and FLOPs. Many implementations also materialize all intermediate basis states, which depresses batch sizes and GPU utilization. Algorithmically, most public code evaluates B-spline bases in time $\mathcal{O}(k \, d_g \, d_{\text{in}} \, d_{\text{out}})$, where $k$ is the B-spline degree, $d_g$ the number of knots (we use "grid" for the sorted knot vector), and $d_{\text{in}}(= n_l)$, $d_{\text{out}}(= n_{l+1})$ the input and output dimensions. Training can also stall when inputs drift outside the initialized knot support, where the B-spline basis vanishes; while Liu et al. (2024b) advocates periodic grid updates, this procedure is numerically brittle and ill-suited to reliable, high-throughput GPU execution.

This work removes these bottlenecks in the original KAN by enabling true batched evaluation and eliminating unnecessary intermediates, substantially reducing both memory footprint and FLOPs.

Table 1: Compute complexity of Torch- and Warp-KAN for single layer B-spline evaluation.

| Model | Complexity |
|---|---|
| Torch KAN | $\mathcal{O}(K d_g d_{in} d_{out})$ |
| Warp KAN | $\mathcal{O}(K d_{in} d_{out})$ |
| Warp UKAN | $\mathcal{O}(K d_{in} d_{out}) + \mathcal{O}_{CG}(d_{emb}^2 + d_{emb} d_{out} K)$ |

Because B-splines also underlie pre-KAN learnable activations, the same kernels accelerate those methods as well.Bohra et al. (2020) Our implementation facilitates a fair comparison between MLPs and KANs; correcting prior apples-to-oranges comparisons that pitted highly optimized MLPs against vanilla implementation of KANsYu et al. (2024). As recent ML systems work has shown (e.g., FlashAttention/FlashFFTConv Dao et al. (2022); Dao (2023); Shah et al. (2024); Fu et al. (2023)), careful software design unlocks scale; we follow the same philosophy for KAN-style architectures. Finally, we introduce UKAN to eliminate grid updates during training and present experiments comparing MLP, KAN, and UKAN across a range of settings.

## 2 ALGORITHM

Our remedy for KAN's compute and memory overhead is to exploit the compact support of B-splines rather than evaluating the Cox–de Boor recurrence over the entire knot vector. A degree-$k$ 1D basis $B_{i,k}$ is nonzero only on $[t_i, t_{i+k+1})$. Leveraging this observation, we represent the 1D B-spline function with basis matrices Qin (1998) as shown in Eq. 8:

$$\text{spline}(u) = U \, \mathbf{M} \, C. \tag{8}$$

Here $x \in \mathbb{R}$ is a scalar input, and we first locate its knot interval via $i = \lfloor (x - t_0)/h \rfloor$, where $h$ is the distance between two adjacent grid points ($h := t_{i+1} - t_i$). We then define the local normalized coordinate

$$u = \frac{x - t_i}{t_{i+1} - t_i} \in [0, 1),$$

so $u$ is a scalar. The vector $U$ is $(1, u, u^2, \ldots, u^k)$, $\mathbf{M}$ is the B-spline basis matrix, and $C$ is the vector of the $k + 1$ local B-spline coefficients, e.g., $(c_i, c_{i+1}, \ldots, c_{i+k})^\top$. The basis matrices are obtained by applying the recursive B-spline equations and depend only on the degree of the B-spline function (see Appendix A.1 for the cubic case).

In the network, Eq. 8 is instantiated for each connection from neuron $i$ in layer $\ell$ to neuron $j$ in layer $\ell + 1$, we evaluate a spline at a scalar argument $u_{i,j}^{(\ell)}$ derived from the activation at neuron $i$,

$$\text{spline}\big(u_{i,j}^{(\ell)}\big) = U\big(u_{i,j}^{(\ell)}\big) \, \mathbf{M} \, C_{i,j}^{(\ell)}.$$

We provide efficient implementations of the above formula using NVIDIA Warp Macklin (2022) in a new library called warpKAN with evaluation complexity of $\mathcal{O}(k d_{in} d_{out})$ along with PyTorch bindings Paszke et al. (2019). This implementation offers both memory and computational efficiency, as described in Table 1. However, the reduction in compute and memory cost of B-spline components does not solve the issue of the bounded range of the grid in the original KAN.

We achieve an unbounded grid by generating B-spline coefficients with a coefficient-generator (CG) MLP, akin to Hyena's filter generation Poli et al. (2023). For a degree-$k$ spline, each evaluation needs $K = k + 1$ coefficients (Eq. 8). A naïve approach—calling the same MLP $K$ times for the $K$ adjacent grid indices performed poorly in our experiments, likely due to missing joint conditioning across those $K$ coefficients. Instead, we partition the uniform grid $\{t_j = j\,h \ : \ \forall j \in \{..., -2, -1, 0, 1, 2, ...\}\}$ into groups of $K$ consecutive cells. For any $x$, define the cell index $i(x) = \lfloor x/h \rfloor$, the group index

$$g = \left\lfloor \frac{i(x)}{K} \right\rfloor = \left\lfloor \frac{x}{K\,h} \right\rfloor, \tag{9}$$

and define the within-group offset $r = i(x) \bmod K$. The CG-MLP takes as input the concatenation of (i) an embedding of the feature index and (ii) a sinusoidal positional encoding of the group index

$g$ (as in Transformers Vaswani et al. (2017)), and outputs a vector $C_g \in \mathbb{R}^K$ of coefficients for group $g$. To ensure the correct sliding window across group boundaries, we concatenate the previous and current outputs $[C_{g-1}; C_g] \in \mathbb{R}^{2K}$ and select the $K$ coefficients

$$\big[C_{g-1}; C_g\big]_{r:\, r+K-1}, \tag{10}$$

which are then used in the basis-matrix evaluation of Eq. 8. This grouping yields an unbounded, index-conditioned parameterization with fewer MLP calls, while preserving the exact $K$-wide local stencil required by B-splines. The general architecture of UKAN is shown in Figure 1. To obtain an interpretable UKAN, one can further decouple the CG-MLP per-edge and add entropy and magnitude regularization terms on the CG-MLP coefficients to the task loss, in the same spirit as the original KAN paper Liu et al. (2024b). A representative pruned graph obtained in this way is shown in Appendix A.2. For inference purposes, one can materialize all coefficients on any finite interval and store them as KAN parameters to avoid repetitive calls to the CG-MLP, since there is a one-to-one mapping between KAN and UKAN (see Appendix A.3). Since at current form MLP compute makes UKAN 3-4x slower than KAN, there is feasibility of fusing CG-MLP and B-spline evaluation into a single kernel but we leave it as a future extension of warpKAN and focus on introducing UKAN and testing it in different tasks.

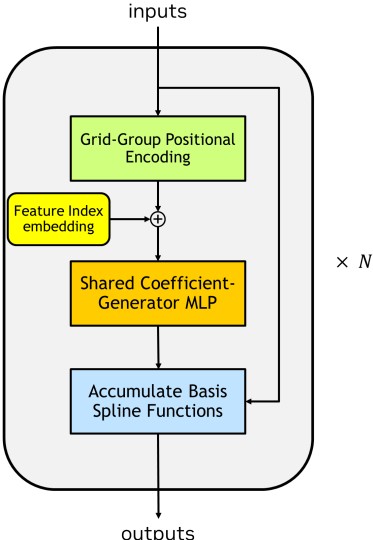

Figure 1: The UKAN model architecture including grid group positional encoding, coefficient-generator MLP, and B-spline function.

## 3 EXPERIMENTS

In this section, we benchmark the performance of warpKAN for various B-spline orders and grid sizes. Subsequently, we showcase the capabilities of UKAN across a wide-range of tasks commonly encountered in machine learning: classification, regression, approximation, generation, and a real-world application to drug discovery.

### 3.1 PERFORMANCE BENCHMARKING

In Figure 2, we benchmark warpKAN against torchKAN under two settings: (a) varying the B-spline order and (b) varying the number of knots ("grid size"). We report the speedup for the forward and backward passes, as well as their sum, for a single layer $KAN(32 \rightarrow 32)$. For panel (a), we use a grid size of 64 and a batch size of $2^{16}$; for panel (b), we fix the B-spline order to 3 and use a batch size of $2^{17}$. All warpKAN results are normalized to the PyTorch implementation from the original paper. Across B-spline orders, warpKAN is $5.5\times - 15\times$ faster, with speedup increasing with

the order. Across grid sizes, warpKAN averages about $12\times$ and reaches up to $24\times$ speedup for larger grids. Moreover, torchKAN runs out of memory for grids $\geq 256$, while warpKAN scales to grid sizes up to $2^{18}$; over $1000\times$ larger. **Notation:** $\text{Layer}(a \rightarrow b)$ denotes input and output dimensions.

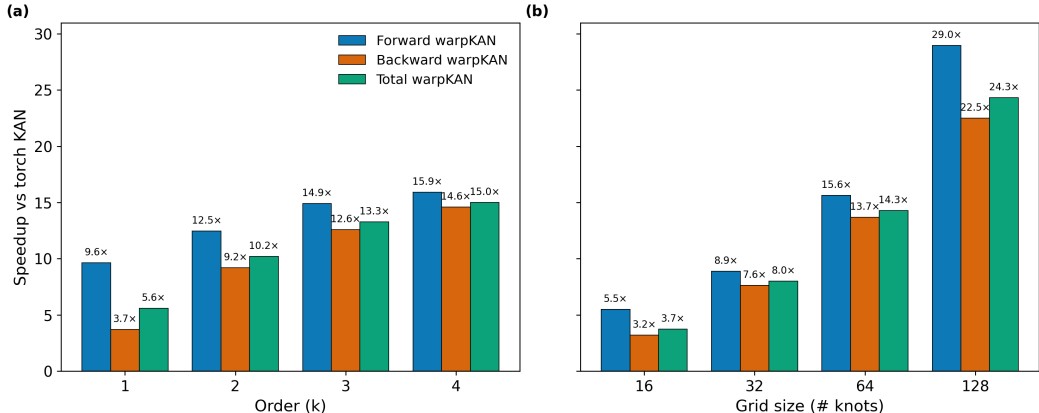

Figure 2: warpKAN vs. torchKAN. (a) Increasing B-spline order yields larger speedups ($5.5\times$–$15\times$). (b) Increasing grid size shows average $\sim 12\times$ and up to $24\times$ speedup; torchKAN hits OOM $\geq 256$ knots while warpKAN scales to $2^{18}$. All values are normalized to the public PyTorch implementation.

From Figure 2, the naïve PyTorch implementation clearly under-utilizes the GPU. Further gains are expected by (i) exploiting Tensor Cores, (ii) fusing kernels to cut launch overhead and off-chip traffic, and (iii) other common techniques in CUDA programming. These optimizations should shift the workload toward the roofline limits and substantially increase throughput, mirroring the trajectory of the FlashAttention family Dao et al. (2022); Dao (2023); Shah et al. (2024).

We quantify how close our implementation is to the hardware limits using a roofline model. The Speed-of-Light (SoL) time, $t_{\text{SoL}}$, corresponds to the limiting factor between compute (FLOP/s) and memory bandwidth (B/s), and thus represents the best achievable runtime on the hardware. Based on this definition, we construct compute and memory models of the B-spline below.

Table 2: B-spline forward SoL and runtime vs. grid size on A6000 (FP32). $t_{meas}$ is our empirical runtime, the remaining columns are theoretical upper and lower bounds. Setup: $N=2^{17}$, $d_{\text{in}}=d_{\text{out}}=32$, order $k=3$ ($K=4$), $P_{\text{FLOP}}=38.7$ TFLOP/s, $P_{\text{BW}}=768$ GB/s.

| $G$ | $t_{\text{meas}}$ (ms) | $t_{\text{SoL}}^{\text{best}}$ (ms) | $t_{\text{SoL}}^{\text{worst}}$ (ms) | $t_{\text{compute}}$ (ms) | $t_{\text{mem}}^{\text{best}}$ (ms) | $t_{\text{mem}}^{\text{worst}}$ (ms) |
|---|---|---|---|---|---|---|
| 32 | 4.9489 | 0.1387 | 2.8399 | 0.1387 | 0.0439 | 2.8399 |
| 64 | 5.1764 | 0.1387 | 2.8399 | 0.1387 | 0.0441 | 2.8399 |
| 128 | 5.3669 | 0.1387 | 2.8399 | 0.1387 | 0.0444 | 2.8399 |
| 256 | 5.5305 | 0.1387 | 2.8399 | 0.1387 | 0.0451 | 2.8399 |
| 512 | 5.7400 | 0.1387 | 2.8399 | 0.1387 | 0.0464 | 2.8399 |
| 1024 | 5.9188 | 0.1387 | 2.8399 | 0.1387 | 0.0492 | 2.8399 |

**Compute model** Let $N$ be the batch size, $d_{\text{in}}$ and $d_{\text{out}}$ the input/output dims, $k$ the B-spline order, $K = k+1$, and $G$ the number of grid positions (knots) per $(i, o)$ pair.[1] We write the per-(input,output) multiply–add count as a $k$-dependent constant $c_k$; in our implementation we use $c_k \approx 2K(K+1)$ (the factor 2 accounts for fused multiply–adds counted as two FLOPs). Then the total work is

$$F_{\text{fwd}} \approx N \cdot d_{\text{in}} \cdot d_{\text{out}} \cdot c_k. \tag{11}$$

**Memory models** We separate optimistic (best-case) and pessimistic (worst-case) coefficient reuse (no-cache and cached). We assume contiguous reads of inputs and writes of outputs; coefficients are indexed locally ($K$ per sample) but stored globally ($(G+K)$ per $(i, o)$, including padding).

---

[1]Our kernels use a local matrix form; $G$ denotes the global grid size before local selection.

*Best-case reuse* (stream all unique coefficients once across the batch):

$$B_{\text{best}} \approx s\left[N\left(d_{\text{in}} + d_{\text{out}}\right) + d_{\text{in}}d_{\text{out}}\left(G+K\right)\right], \qquad s = 4 \text{ bytes (FP32)}. \tag{12}$$

*Worst-case reuse* (fetch local $K$ coefficients per sample):

$$B_{\text{worst}} \approx s\left[N\left(d_{\text{in}} + d_{\text{out}}\right) + N\,d_{\text{in}}d_{\text{out}}\,K\right]. \tag{13}$$

The SoL times are then

$$t_{\text{SoL}}^{\text{best}} = \max\left(\frac{F_{\text{fwd}}}{P_{\text{FLOP}}}, \frac{B_{\text{best}}}{P_{\text{BW}}}\right), \qquad t_{\text{SoL}}^{\text{worst}} = \max\left(\frac{F_{\text{fwd}}}{P_{\text{FLOP}}}, \frac{B_{\text{worst}}}{P_{\text{BW}}}\right). \tag{14}$$

We perform experiments for different configurations on the A6000 GPU in FP32, and report results in Table2. We time forward calls with CUDA synchronization before and after, discarding warm-up. We use fixed shapes $(N, d_{\text{in}}, d_{\text{out}}, k) = (2^{17}, 32, 32, 3)$, report the average per-call time, and compute $F_{\text{fwd}}$ via equation 11 with $c_k=2K(K+1)$. We report both $t_{\text{SoL}}^{\text{best}}$ and $t_{\text{SoL}}^{\text{worst}}$ using equation 12–equation 13. As shown in Table 2, there are still opportunities to improve runtime of B-spline used in KANs and many other workloads. The future work will try to close the gap between runtime and SoL time.[2] The runtime of vanilla KAN implementations can be estimated by multiplying the speedup values reported in Figure 2 with the measured $t_{\text{meas}}$ from Table 2, provided the configuration does not run out of memory.

We also study the memory and compute cost of UKAN relative to KAN, where both implementations use the accelerated warpKAN backend rather than the naïve torch implementation. For a single layer with identical input/output dimension, B-spline order 3, and grid size 32, UKAN incurs approximately $3.7\times$ higher runtime and about $210\times$ larger GPU memory footprint than KAN, because UKAN must generate and store per-edge coefficients via the CG-MLP whereas KAN reuses a fixed table of B-spline coefficients. The resulting comparison is summarized in Figure 3. We also provide possible solution to the memory and compute in Appendix **??**, where we provide prototype experimental code for UKAN with better performance and memory fingerprints, where UKAN consumes less memory compared to the KAN as B-spline coefficients are not materialized.

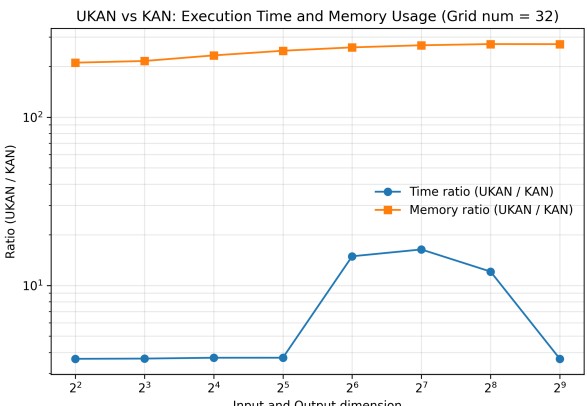

Figure 3: Compute and memory comparison of KAN vs. UKAN, both implemented with warp-KAN. We report wall-clock time and peak GPU memory for a single layer with order-3 B-splines and grid size 32 on an NVIDIA A6000 (batch size 1024). All points are normalized so that KAN= 1.0; in this setting UKAN is roughly $3.7\times$ slower and $210\times$ more memory hungry due to the cost of generating and materializing per-edge coefficients from the CG-MLP.

---

[2]We do not provide a full roofline breakdown for UKAN, as its cost is dominated by the CG network. Our ongoing work explores a fused implementation that co-schedules the CG MLP with spline evaluation and accumulation in a single kernel; so that coefficients are produced and consumed without intermediate HBM round-trips. These experiments are in progress.

## 3.2 TASKS

### 3.2.1 REGRESSION

To evaluate the accuracy of our UKAN and KAN in regression, we conducted three experiments.

    I. $f(x, y) = \exp(J_0(20x) + y^2)$, where $J_0$ is the Bessel function of order 0.

    II. $f(x, y) = \exp(\sin \pi x + y^2)$

    III. $f(\mathbf{x}) = \exp(\frac{1}{15} \sum_i^n \sin((\frac{4i}{15} + 1)\pi x_i))$ , where $i = 0, 1, \ldots, 15$ and is a high dimensional function compared with functions I and II.

We compare the results of UKAN, KAN and MLP $(2 \to 5 \to 1)$ for 2D functions (functions I and II). For function III, we use UKAN, KAN, and MLP $(16 \to 32 \to 1)$. For the CG model, we used a two-layer MLP with 8- and 16-dimensional positional encodings and feature embeddings for 2D and 16D functions, respectively. The first layer of the CG MLP uses SiLU nonlinearity and generated coefficients are scaled by another learnable parameter to improve learning, analogous to the original KAN paper. An Adam optimizer Kingma & Ba (2014) with a learning rate of 0.01 and weight decay of $1e^{-5}$ for 200,000 epochs is used to minimize the MSE loss. The learning rate is decayed exponentially with the rate of $1 - 1e^{-4}$ and minimum learning rate of $1e^{-4}$. The results are shown in Figure 4, where UKAN and KAN perform much better than MLP, and KAN performs better than UKAN. In theory, KAN and UKAN have the same learning capacity, but the MLP component of UKAN might slightly hurt generalization and performance compared to KAN. We also note that although the compute cost of a KAN is greater than that of an MLP by a factor of $K$, KAN convergence often compensates for this factor. We also performed a regression task on the n-body problem based on Satorras et al. (2022), the results are available in Appendix A.4.

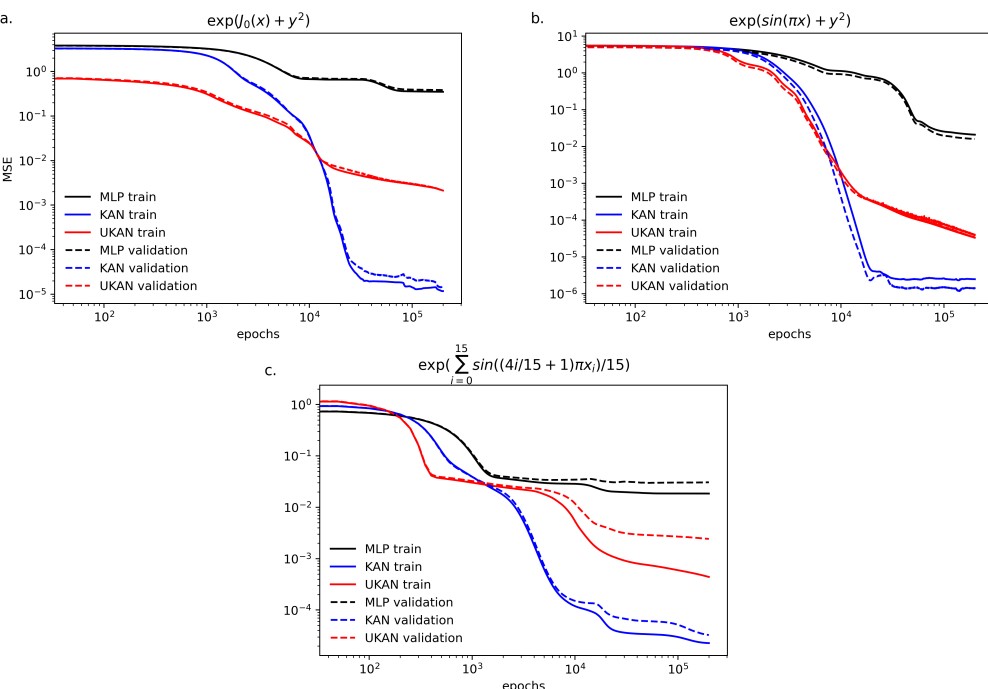

Figure 4: Regression task results. (a) RMSE vs. training epochs for Function I using KAN, UKAN, and MLP$(2 \to 5 \to 1)$. (b) RMSE vs. training epochs for Function II using KAN, UKAN, and MLP$(2 \to 5 \to 1)$. (c) RMSE vs. training epochs for Function III using KAN, UKAN, and MLP$(16 \to 32 \to 1)$.

### 3.2.2 CLASSIFICATION

We devised four experiments to evaluate KAN and UKAN on classification tasks. Two of these use an MLPMixer architecture with KAN and UKAN backbones on CIFAR-10 and ImageNet-1K Tolstikhin et al. (2021); Krizhevsky et al. (2009); Deng et al. (2009); their details are provided in Appendix E. Here, we focus on the remaining two experiments on smaller datasets, where we train UKAN and KAN on the Moon and MNIST datasets. The training and validation accuracies on the Moon dataset are reported in Table 3. We observe that both models achieve close to $100\%$ accuracy on this 2D task, with UKAN showing slightly higher accuracy than KAN in this setup. For this experiment, both UKAN and KAN $(2 \rightarrow 4 \rightarrow 2)$ are trained using stochastic gradient descent (SGD) with a learning rate of 0.01 for 10,000 epochs, and results are averaged over three different random initializations.

Table 3: Moon dataset classification accuracy.

| Model | Training | Validation |
|-------|----------|------------|
| KAN | $98.46 \pm 1.3$ | $98.53 \pm 0.4$ |
| UKAN | $\mathbf{100.0 \pm 0}$. | $\mathbf{99.83 \pm 0.17}$ |

The final classification task was performed on the MNIST dataset, where we trained both the UKAN and KAN models with configurations $(784 \rightarrow 32 \rightarrow 10)$ and a degree-3 B-spline. Both models were optimized using the Adam optimizer combined with an Exponential scheduler, having a learning rate of $2 \times 10^{-4}$ and a decay rate of $1 - 10^{-4}$. The KAN network incorporated 51 grid points across the interval $[-10, 10]$, whereas UKAN utilized a grid delta of 3.0 and a 48-dimensional positional encoding. Notably, both models employed only the B-spline component without any MLP components. Training was halted upon detection of overfitting in the training dataset. Furthermore, three rounds of independent training with different initializations were conducted to compare the performance of UKAN and KAN. The results, as presented in Table 4, indicate that UKAN outperforms KAN on the validation dataset while slightly underperforming on the training dataset.

Table 4: MNIST dataset classification accuracy.

| Model | Training | Validation |
|-------|----------|------------|
| KAN | $\mathbf{98.93 \pm 0.78}$ | $95.35 \pm 0.04$ |
| UKAN | $98.40 \pm 0.24$ | $\mathbf{96.29 \pm 0.08}$ |

### 3.2.3 APPROXIMATION

We explore the effectiveness of UKAN and KAN in physics-informed neural networks Karniadakis et al. (2021) to solve the logistic growth model, which is used to model population dynamics in biological and ecological systems. For this experiment, we use both UKAN and KAN $(1 \rightarrow 5 \rightarrow 1)$ without MLP component to solve the differential equation below,

$$\frac{df}{dt} = Rf(t)(1 - f(t)) \tag{15}$$

where $R$ is the growth rate set to 1.0 and the function $f(t)$ represents the growth rate of the population over time (t). We impose boundary condition of $f(0) = 0.5$ to uniquely specify the solution and compare the results with the analytical solution of $f(t) = \frac{1}{1+\exp(-t)}$. We use domain of $[-5, 5]$ to sample data and Adam optimizer with the learning of rate of $1e^{-3}$ and weight decay of $1e^{-5}$ and follow the standard procedure for PINN minimization, i.e. minimizing MSE of the differential equation residual over collocation points and boundary conditions. The results are shown in Figure 5, for and KAN $(1 \rightarrow 5 \rightarrow 1)$ without the MLP component. UKAN and KAN achieve MSE of $1e^{-5}$ and $1e^{-6}$, respectively on the sample dataset, indicating both models are applicable to physics-informed neural networks scenarios.

### 3.2.4 GENERATION

As an example in generative model learning, we evaluated the performance of three different architectures for Denoising Diffusion Probabilistic Models (DDPM) Ho et al. (2020) on a synthetic 2D

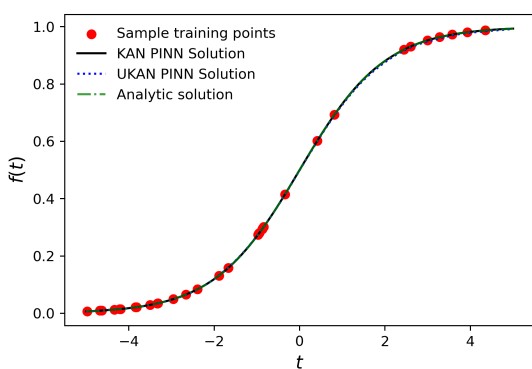

Figure 5: KAN and UKAN used in PINNs. Solving logistic growth model with both KAN and UKAN ($1 \to 5 \to 1$) over domain of -5 and 5.

circle dataset with added noise. The first architecture is only composed of MLPs, while other architectures use KAN and UKAN in the input of temporal layers and output layer (see Appendix A.5 for full architecture). We used Adam optimizer with a learning rate of $5e^{-5}$ for 500 epochs with a batch size of 800. Our results demonstrated that both KAN and UKAN significantly outperformed MLP in terms of the Wasserstein distance shown in Table 5 and sample quality as shown in Figure 6. Data samples from original distribution and generated from DDPM with KAN, UKAN, and MLP architectures indicates superior performance of KAN and UKAN compared to MLP and slightly superior performance of UKAN over KAN. This result indicates possible applications of KAN and UKAN in generative tasks, where MLP alone might fail to learn underlying data distribution especially in sample quality as we observed loss values of MLP, KAN and UKAN were very small.

Table 5: DDPM with KAN, UKAN, and MLP

| Model | Wasserstein distance |
|-------|----------------------|
| KAN   | 0.693                |
| UKAN  | **0.655**            |
| MLP   | 1.058                |

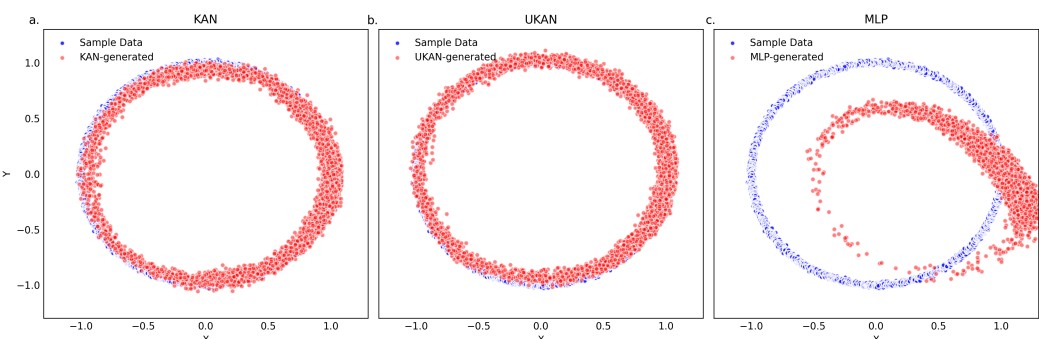

Figure 6: DDPM with KAN, UKAN, and MLP.

### 3.2.5 REAL-WORLD APPLICATION: DRUG DISCOVERY

Accurate *in silico* prediction of molecular properties is key to accelerating drug discovery by early identification of compounds with favorable ADME (Absorption, Distribution, Metabolism, and Excretion) profiles. Ferreira & Andricopulo (2019); Beckers et al. (2023); Seal et al. (2025) Machine learning methods have shown strong performance in predicting these properties from chemical struc-

ture or computed features. The usage of KANs in predicting molecular properties was originally introduced by Li et al. (2025). Due to implementation discrepancies and known issues within the MoleculeNet datasets,Walters we reimplemented the original model using the warpKAN package and incorporated the necessary corrections. A comprehensive analysis of these results is presented in the Appendix C.2.

Here, we further demonstrate the ability of UKANs to predict ADME properties, and used a fixed molecular representation to isolate the impact of the KAN and UKAN addition. We choose Morgan fingerprints and RDKit 2D descriptors as our two computed molecular features. In order to overcome the limitations of the MoleculeNet dataset, we use four datasets released by Fang et al. (2023). These models were trained for 200 epochs using Adam optimizer with a learning rate of 1e-4. As shown in Table 6, UKANs show superior or comparable performance with respect to KANs for predicting molecular properties from computed features using mean absolute error metric while containing two orders of magnitude fewer parameters. UKANs outperform KANs on the permeability and solubility dataset using Morgan fingerprint features while their performance is statistically equivalent (using T-test for means) on the other two datasets. Using RDKit 2D descriptors, a set of 200 molecular descriptors describing the 2D structure of the molecule, UKAN shows better performance than KAN on the Microsomal stability (rat) dataset. These results show that UKANs are superior to KANs in accurately predicting molecular properties relevant to the pharmaceutical industry. Details about model architecture and data preprocessing can be found in the Appendix A.6 and B.1 respectively. More details on the robustness of predictions as a function of distance from the training set are discussed in C.1.

Table 6: Performance of KAN and UKAN on Biogen ADME datasets using Morgan fingerprint (labeled Morgan) and RDKit 2D descriptors (labeled RDKit-2D) as molecule featurizers. Performance is assessed using mean absolute error on test split as the metric. ($\downarrow$ is better)

|  | Descriptor | Microsomal stability (human) | Microsomal stability(rat) | Permeability | Solubility |
|---|---|---|---|---|---|
| KAN | Morgan | $0.479 \pm 0.010$ | $0.559 \pm 0.012$ | $0.424 \pm 0.005$ | $0.476 \pm 0.006$ |
| UKAN | Morgan | $0.476 \pm 0.003$ | $0.551 \pm 0.020$ | $\mathbf{0.403 \pm 0.006}$ | $\mathbf{0.435 \pm 0.009}$ |
| KAN | RDKit-2D | $0.373 \pm 0.003$ | $0.492 \pm 0.010$ | $0.349 \pm 0.005$ | $0.386 \pm 0.002$ |
| UKAN | RDKit-2D | $0.368 \pm 0.008$ | $\mathbf{0.467 \pm 0.006}$ | $0.348 \pm 0.003$ | $0.397 \pm 0.013$ |

## 4 CONCLUSION

In this work, we presented the Unbound Kolmogorov-Arnold Network (UKAN), which unifies multilayer perceptron networks (MLPs) with KANs along with an efficient GPU implementation of the underlying components of KANs. GPU acceleration decouples the computational cost and memory fingerprint of KANs from the grid size by using local matrix representations of B-spline functions. In addition, our proposed UKAN architecture allows the use of KANs without any fixed grid range limitation by generating coefficients from a coefficient-generator MLP. We evaluated UKAN model for regression, classification, and generative tasks. Our accompanying GPU library alleviates the core bottlenecks that have limited the practical scale of spline-based networks and serves as a reusable building block for future models. We expect UKAN and its variants to enable accelerated large-scale learning in domains such as molecular property prediction, protein docking, language, and vision, and we see promising directions in multi-GPU training, adaptive knot policies, and a deeper theory of approximation on unbounded domains.

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

## A APPENDIX

You may include other additional sections here.

### A.1 CUBIC B-SPLINE BASIS MATRIX REPRESENTATION

For a B-spline of order 3, the basis matrix representation can be written as,

$$
\text{spline}(u) = \begin{bmatrix} 1 & u & u^2 & u^3 \end{bmatrix} \frac{1}{6} \begin{bmatrix} 1 & 4 & 1 & 0 \\ -3 & 0 & 3 & 0 \\ 3 & -6 & 3 & 0 \\ -1 & 3 & -3 & 1 \end{bmatrix} \begin{bmatrix} c_0 \\ c_1 \\ c_2 \\ c_3 \end{bmatrix}.
$$

### A.2 INTERPRETABLE UKAN

A key motivation behind KANs is that they expose an interpretable computation graph in which each edge carries a low-dimensional spline whose coefficients can be inspected, pruned, or even symbolically regressed. Although UKAN replaces the fixed spline table with a coefficient-generator MLP (CG-MLP), we can recover a similarly interpretable representation by slightly modifying the training procedure. Here, we describe how to recover an interpretable representation in UKAN by

(i) decoupling the coefficient-generator MLP (CG-MLP) on a per-edge basis and (ii) adding L1 (magnitude) and entropy regularization terms on the generated B-spline coefficients, analogously to the original KAN formulation Liu et al. (2024b).

For each edge $(i \to j)$ in a given layer $\ell$, the associated CG-MLP outputs a vector of B-spline coefficients

$$c_{i,j}^{(\ell)} \in \mathbb{R}^K,$$

where $K$ is the number of spline basis functions on that edge. These coefficients replace the fixed KAN table entries and are used in the basis-matrix evaluation described in Eq. 8.

**L1 (magnitude) regularization.**  Following Liu et al. (2024b), we define an L1 magnitude for each edge as

$$\left\| c_{i,j}^{(\ell)} \right\|_1 = \sum_{k=1}^{K} \left| c_{i,j,k}^{(\ell)} \right|, \tag{16}$$

and the L1 magnitude of layer $\ell$ as the sum over all edges in that layer,

$$\left\| \Gamma_\ell \right\|_1 = \sum_{i=1}^{n_{\mathrm{in}}^{(\ell)}} \sum_{j=1}^{n_{\mathrm{out}}^{(\ell)}} \left\| c_{i,j}^{(\ell)} \right\|_1, \tag{17}$$

where $\Gamma_\ell$ denotes the collection of all CG-MLP coefficients in layer $\ell$.

**Entropy regularization.**  To encourage each edge to concentrate its mass on a small subset of spline segments, we normalize the absolute coefficients on edge $(i \to j)$:

$$p_{i,j,k}^{(\ell)} = \frac{\left| c_{i,j,k}^{(\ell)} \right|}{\left\| c_{i,j}^{(\ell)} \right\|_1 + \varepsilon}, \qquad k = 1, \ldots, K, \tag{18}$$

with a small $\varepsilon > 0$ added for numerical stability. The entropy of that edge is then

$$S\big(c_{i,j}^{(\ell)}\big) = -\sum_{k=1}^{K} p_{i,j,k}^{(\ell)} \log p_{i,j,k}^{(\ell)}, \tag{19}$$

and the entropy of layer $\ell$ is

$$S(\Gamma_\ell) = \sum_{i=1}^{n_{\mathrm{in}}^{(\ell)}} \sum_{j=1}^{n_{\mathrm{out}}^{(\ell)}} S\big(c_{i,j}^{(\ell)}\big). \tag{20}$$

**Total loss.**  Let $\ell_{\mathrm{pred}}$ denote the task (prediction) loss. The total training objective for UKAN with interpretability regularization is

$$\ell_{\mathrm{total}} = \ell_{\mathrm{pred}} + \lambda \left( \mu_1 \sum_{\ell=0}^{L-1} \left\| \Gamma_\ell \right\|_1 + \mu_2 \sum_{\ell=0}^{L-1} S(\Gamma_\ell) \right), \tag{21}$$

where $\lambda$ is a global regularization weight and $(\mu_1, \mu_2)$ balance the relative strength of the L1 and entropy terms, respectively. This is a direct analog of the KAN objective (Eqs. (2.17)–(2.20) in Liu et al. (2024b)), with activation functions replaced by CG-MLP–generated coefficient vectors.

**Pruning and visualization.**  After training, we compute the effective contribution of each edge by combining its learned scalar weight (if present) and the magnitude of its coefficient vector $c_{i,j}^{(\ell)}$. Edges and spline segments whose contribution falls below a fixed threshold are pruned. The resulting sparse computation graph is then visualized in the same manner as in the original KAN paper.

For the regression task

$$y = \exp\big(\sin(\pi x_1) + x_2^2\big),$$

The pruned UKAN graph obtained in this setting (Figure 7 in the appendix) recovers the expected quadratic dependence on $x_2$ and sinusoidal dependence on $x_1$, and matches the original KAN pruned graph up to a permutation of hidden nodes.

756
757
758
759
760
761
762
763
764
765
766
767
768
769
770
771
772
773
774
775
776
777
778
779
780
781
782
783
784
785
786
787
788
789
790
791
792

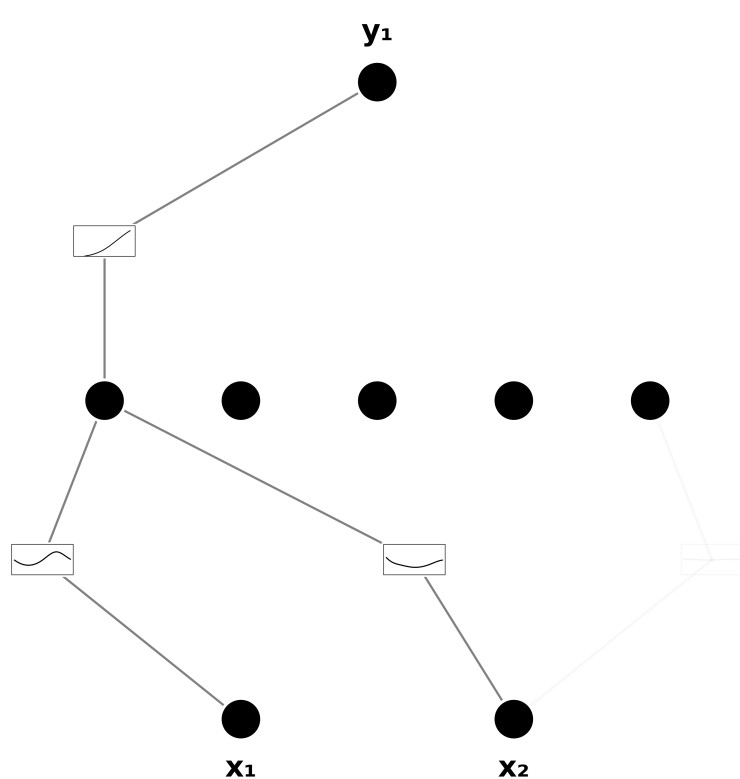

Figure 7: Pruned UKAN model for the regression task $y = \exp\big(\sin(\pi x_1) + x_2^2\big)$.

### A.3 UKAN MAPPING TO KAN IN INFERENCE

**Proposition (UKAN $\rightarrow$ KAN at inference).** Fix B–spline degree $k$ and uniform grid spacing $h$, and set $K := k + 1$. Consider a UKAN whose coefficient–generator (CG) produces a group vector $G_g \in \mathbb{R}^K$ for each group index $g \in \mathbb{Z}$. Evaluation uses the same local stencil as KAN,

$$\text{spline}(x) = U(u)\, M_k\,(\cdot), \qquad U(u) = (1, u, \ldots, u^k), \;\; u = \tfrac{x - t_i}{h} \in [0, 1),$$

with $t_i = i\,h$ the knot at cell index $i$. Let $\mathcal{X} \subset \mathbb{R}$ be the (finite) set of inputs on which inference is performed; more generally, let $\mathcal{I}$ be any compact interval containing $\mathcal{X}$. Then there exists a KAN with coefficients $\{C_i\}$ defined only for cells intersecting $\mathcal{I}$ such that the KAN exactly matches the UKAN on $\mathcal{X}$ (indeed, on all of $\mathcal{I}$).

*Proof.* For each cell index $i$ whose cell $[t_i, t_{i+1})$ intersects $\mathcal{I}$, write $i = gK + r$ with

$$g = \left\lfloor \tfrac{i}{K} \right\rfloor, \qquad r = i \bmod K \in \{0, \ldots, K - 1\}.$$

Query the CG once (in inference mode; parameters fixed) to obtain the adjacent group vectors $G_{g-1}, G_g \in \mathbb{R}^K$, and *materialize* the KAN's local coefficients by the sliding window

$$C_i \;:=\; \text{slice}\big([G_{g-1}; G_g], \; r : r + K\,\big) \in \mathbb{R}^K,$$

where $[\,\cdot\,;\,\cdot\,]$ denotes concatenation and $\mathrm{slice}(\cdot, a{:}b)$ extracts entries $a, \ldots, b-1$. By construction, for any $x$ in cell $i$ the UKAN evaluation applies the same linear stencil $U(u)M_k$ to the same length-$K$ window drawn from $[G_{g-1}; G_g]$. Hence

$$U(u)M_k\, C_i \;=\; U(u)M_k\, \mathrm{slice}(\,[G_{g-1}; G_g],\, r{:}r + K\,),$$

which is exactly the UKAN output in that cell. Because $\mathcal{I}$ intersects finitely many cells, the above defines finitely many $C_i$ and therefore a finite–parameter KAN that agrees with the UKAN on all $x \in \mathcal{I}$, in particular on the finite inference set $\mathcal{X}$.

## A.4  N-BODY PROBLEM

KANs promise better generalization compared to MLPs for regression tasks, similar to equivariant models allowing for the exploitation of symmetries for improved generalization. In particular, E(n)-Equivariant Graph Neural Networks (EGNNs) are equivariant with respect to the translations, rotations, and permutations Satorras et al. (2022). Here, we explore how combining equivariance with KAN leads to improved performance in the study of n-body systems as described in the EGNN paper Satorras et al. (2022). To evaluate this, we replace the final scalar predicting MLPs in EGNN with UKAN and KAN layers. Specifically, the scalar outputs of $\phi_x$ and $\phi_v$ in equations below are predicted with UKAN and KAN.

$$\mathbf{v}_i^{\ell+1} = \phi_v(\mathbf{h}_i^\ell)\, \mathbf{v}_i^{\mathrm{init}} + C \sum_{j \neq i} (\mathbf{x}_i^\ell - \mathbf{x}_j^\ell)\, \phi_x(\mathbf{m}_{ij}^{\ell+1}), \qquad \mathbf{x}_i^{\ell+1} = \mathbf{x}_i^\ell + \mathbf{v}_i^{\ell+1}.$$

Where $\mathbf{x}_i^{(\ell)}$ and $\mathbf{v}_i^{(\ell)}$ are the position and velocity of $i^{th}$ particle in the $l^{th}$ layer of EGNN. We keep the rest of parameters and datasets identical to the original paper and their code on Github. We also train the SE(3) Transformer model as another reference point. The results are shown in Table 7, where we observe that UKAN and KAN improve the accuracy compared to the original architecture, and the improvement of UKAN is better than the KAN model.

Table 7: Mean Squared Error for the future position prediction in the N-body system.

| Method | MSE |
|---|---|
| EGNN | 0.00638 |
| EGNN+KAN | 0.00609 |
| EGNN+UKAN | **0.00591** |
| SE(3) Transformer | 0.02469 |

## A.5  ARCHITECTURE DETAILS OF DDPM

The Decoder network is designed to transform input features through a series of linear and temporal layers. Here we explain architecture without KAN or UKAN layers, *i.e.* with only linear layer and SiLU nonlinearity, and mention the differences at the end. The architecture consists of an input linear layer, three temporal layers, and an output linear layer.

The Decoder is constructed with the following layers:

- **Input Linear Layer:** The initial fully connected layer transforms the input features from the input dimension to an intermediate dimension.

- **Temporal Layers:** A series of temporal layers; specifically designed for the handling of time-dependent data. In our implementation, we use three temporal layers.

- **Output Linear Layer:** The final fully connected layer transforms the intermediate features back to the original input dimension.

- **Nonlinearity:** The intermediate features passed through SiLU non-linear activation function before being processed by the temporal layers.

The Temporal Layer is designed to integrate temporal information into the feature transformation process. This layer receives the input features and a temporal embedding, processes them through a series of linear transformations, and combines the outputs with a skip connection to ensure that the temporal information is effectively incorporated.

The Temporal Layer consists of the following components:

- **Fully Connected Layers:** These layers perform linear transformations on the input features.
- **Temporal Encoding:** This layer projects the temporal embedding to the same dimensional space as the output features.
- **Skip Connection:** If the input and output features have the same dimension, an identity mapping is used. Otherwise, a linear transformation is applied to match the dimensions.
- **Output Linear Layer:** This layer produces the final output by combining the transformed features with the skip connection.

Within KAN and UKAN architectures, we only replaced the output linear layers of the Decoder network and Temporal layers with UKAN and KAN layers. We used UKAN with grid delta of 0.4 and 24 dimensional positional encoding and KAN with 11 grid between -2 and 2. Both KAN and UKAN were order 3 B-spline functions without MLP component.

### A.6 ARCHITECTURE DETAILS OF MOLECULAR PROPERTY PREDICTION

The model architecture consists of 3 layers of KAN/UKAN with hidden dimension equal to 2X the dimension of the input computed molecular features. The Morgan fingerprint was computed using RDKit packageLandrum et al. (2016) with a radius of 2 and a bit length of 1024, yielding a molecular features of dimension 1024. RDKit 2D descriptors were computed using RDKit package and normalized using descriptastorus packagedes resulting in a vector of size 200. Both KAN and UKANs were used with order 3 B-spline functions and a grid delta of 1.0. KANs usually contains far higher number of parameters in comparison to UKANs. In this setting, UKAN models contain 2 orders of magnitude fewer parameters but still show superior accuracy in comparison to KANs.

Table 8: Number of parameters of KAN and UKAN models with computed molecular features

| Descriptor | KAN | UKAN |
|---|---|---|
| Morgan fingerprint | 2.56B | 6.66M |
| RDKit descriptors | 97.84M | 312K |

### A.7 ARCHITECTURE DETAILS OF GCN

**KAN/UKAN/FKAN-GCN** We build three GCN variants by replacing MLP block (node embedding, message passing, and readouts) with a KAN, UKAN, or FKAN layer, respectively.

**Node embedding.** Given initial node features $f_v \in \mathbb{R}^{d_{in}}$ and neighbor set $\mathcal{N}(v)$, we form an embedding by concatenating $f_v$ with a degree-normalized neighbor average and passing it through a basis layer $\Phi_E(\cdot)$ implemented by KAN/UKAN/FKAN:

$$h_v^{(0)} = \Phi_E\left( \left[ f_v; \frac{1}{|\mathcal{N}(v)|} \sum_{u \in \mathcal{N}(v)} f_u \right] \right)$$

**Message passing.**

At layer $\ell$, the neighbor is first transformed by the basis layer $\Phi_M^\ell(\cdot)$ and aggregated over the neighbor with summation.

$$m_{uv}^{(\ell)} = \Phi_M^{(\ell)}(h_u^{(\ell)}), \qquad h_v^{(\ell+1)} = \sum_{u \in \mathcal{N}(v)} m_{vu}^{(\ell+1)}.$$

**Readout.** After $L$ layers (four layers in our experiments), node features are pooled with AVG (any permutation-invariant $P \in \{\text{AVG}, \text{SUM}, \text{MAX}\}$ is supported) and mapped to outputs by a KAN/UKAN/FKAN readout:

$$\bar{h} = P\big(\{h_v^{(L)} : v \in V\}\big), \qquad \hat{y} = \Phi_R(\bar{h}).$$

Here, each $\Phi_{\{E,M,R\}}(\cdot)$ denotes the same structural layer instantiated with a different basis: KAN uses fixed, bounded B-spline grids; UKAN uses a coefficient-generator to produce local B-spline coefficients on an unbounded symmetric grid; FKAN replaces the spline basis with a Fourier basis. All variants are drop-in compatible.

## B    DATA PREPREPROCESSING

### B.1    MOLECULAR PROPERTY PREDICTION

The molecular property prediction dataset was curated using standard practices in cheminformatics like removal of invalid SMILES, molecule standardization, removal salts, charge neutralization, and mean aggregation of any duplicate labels. This curation practice roughly following the guidelines outlined in Fourches et al. (2010). The molecules were split into train, validation, and test set using Bemis-Murcko scaffold split for each of the tasks in the dataset. In order to make statistically significant comparisons, only tasks containing a more than 500 labels were used. This resulted in selecting 4 out of the 6 tasks presented in the original paper.Fang et al. (2023)

## C    RESULTS

### C.1    ANALYSES OF DRUG DISCOVERY SECTION

The mean absolute error of the prediction of KAN and UKAN on test set molecules binned by the distance to their nearest neighbor in the training set is shown in Figure 8. Results of Morgan fingerprint fixed descriptor are used as it is straightforward to compute distance between two molecules. The distance between two molecules $m_i$ and $m_j$ with Morgan fingerprint $f_i$ and $f_j$ is computed as $d(m_i, m_j) = 1 - \text{TanimotoSimilarity}(f_i, f_j)$. Both KAN and UKAN show similar performance characteristics with error increasing as test set molecules become more dissimilar to the training set. UKAN shows slightly lower error at larger distances to training set indicating more robust generalization to out-of-distribution datasets.

### C.2    KAN/UKAN/FKAN-GCN RESULTS ON MOLECULENET

Li et al. (2025) introduced a graph neural network with multi-layer perceptron layers replaced with KANs. The authors demonstrate that such an architectures shows strong performance on the MoleculeNet datasets. However, we identified two issues with the KAN-GCN implementation by Li et al.: (1) Best model selection is performed on test loss instead of validation loss, which is a deviation from the best practices in machine learning. (2) MoleculeNet consists of multi-task datasets with missing labels. The missing labels were assigned a value of 0.0 instead of being treated as a missing label. Train and test metrics were computed based on this artificial label. We have fixed these issues in our reimplementation using warpKAN library.

To assess scalability of our implementation, we evaluate KAN, UKAN, and Fourier-based KAN (FKAN) graph convolutional neural network (GCN) on seven MoleculeNet datasets Wu et al. (2018). The total number of components in the dataset is over 148,000 molecules, where each molecule is composed of several heavy atoms, e.g., BACE and Tox21 have on average 65 and 36 atoms. While the MoleculeNet datasets have many issues as noted in Walters, we have run this experiments in order to compare directly with the results of KAN-GCN. Additionally, some of the MoleculeNet datasets are also significantly larger than the Biogen ADME datasets. For example, the MUV dataset from MoleculeNet is 30X larger than the largest dataset in Biogen ADME. The number of parameters also varies significantly among the models with UKAN-GNN and FKAN-GCN containing $\sim$75K parameters and $\sim$45K parameters respectively. KAN-GCN models are much larger with $\sim$2.3M parameters.

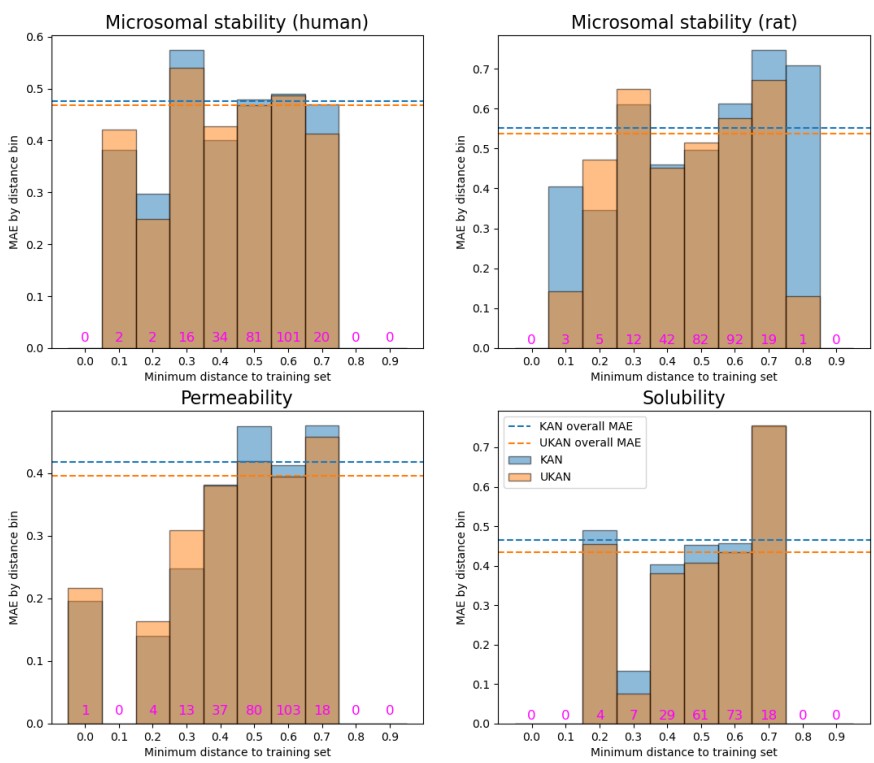

Figure 8: A figures showing the performance of UKAN and KAN using Morgan fingerprint fixed descriptor as a function of distance of test set molecules to their nearest molecule in training set. The number of test set molecules in each bin is shown at the bottom of the bars.

We modify a standard GCN Wang et al. (2023) by replacing its MLP blocks with KAN or UKAN layers (see Appendix A.7). To the best of our knowledge, our training is one of the large-scale training that incorporates KAN in all parts of GCN. Note while works like Zhang & Zhang (2024); Bresson et al. (2025) claim KAN usage it is usually limited to initial node embedding or readouts, as without efficient implementations like ours it is infeasible to investigate KAN in large scale. We train three models, where we divide data into training, validation, and testing datasets and select the best model based on the validation loss. We report the area under the curve (AUC) for models in Table 9. Each model is trained for 500 epochs using Adam optimizer with learning rate of 2e-4 and a step scheduler with a decay rate of 0.9 every 20 steps. Unlike values reported in Li et al. (2025), we see the accuracy of B-Spline- and Fourier-based models are very close to each other, when model selection is done correctly. The statistically best performing model(s) (using T-test for means) are reported in **bold**.

Table 9: Performance of KAN-GCN, UKAN-GCN, and FKAN-GCN on MoleculeNet datasets (↑ is better).

| Dataset | BBBP | BACE | ClinTox | Tox21 | SIDER | HIV | MUV |
|---|---|---|---|---|---|---|---|
| Tasks | 1 | 1 | 2 | 27 | 12 | 1 | 17 |
| # molecules | 2,039 | 1,513 | 1,477 | 7,831 | 1,427 | 41,127 | 93,087 |
| KAN | 65.6(2.2) | 80.0(2.4) | **96.5(0.2)** | **79.4(0.7)** | **83.5(0.2)** | 76.5(1.3) | 71.8(5.7) |
| UKAN | 64.5(2.2) | 74.9(6.3) | 94.7(0.9) | 77.4(0.5) | **83.7(0.2)** | 75.2(1.0) | 69.6(2.1) |
| FKAN | 67.0(2.0) | 80.5(3.0) | **96.5(0.1)** | **79.4(0.8)** | 82.2(0.1) | 74.1(2.2) | **76.5(1.2)** |

## D   FUSED UKAN

Here, we provide a code snippet in warp where we fused CG-MLP with B-spline evaluation in order to reduce memory and runtime of UKAN compared to KAN.

Listing 1: Fused UKAN order-1 forward kernel in Warp

```
EMBED_DIM = 16
K = 1
DIM_IN = 32
DIM_OUT = 32

@wp.func
def silu(x: float) -> float:
    return x / (x.dtype(1.0) + wp.exp(-x))

@wp.kernel
def fused_ukan_kernel_order_1_fwd(x: wp.array2d(dtype=float),
                                  w1: wp.array2d(dtype=float),
                                  w2: wp.array2d(dtype=float),
                                  # TODO: add local embeds
                                  y: wp.array2d(dtype=float)):

    b, tid = wp.tid()
    x_tile = wp.tile_load(x[b], shape=(DIM_IN), offset=(0))

    grid_embeds_cur = wp.tile_zeros(shape=(DIM_IN, EMBED_DIM), dtype=float
        )
    grid_embeds_prev = wp.tile_zeros(shape=(DIM_IN, EMBED_DIM), dtype=
        float)

    for ij in range(tid, wp.static(DIM_IN * EMBED_DIM), 256):
        i = ij // EMBED_DIM
        j = ij % EMBED_DIM
        id_group = wp.float32(wp.int32(wp.floordiv(x_tile[i], wp.float32
            (0.2))) / (wp.static(K) + 1))

        grid_embeds_cur[i, j] = wp.sin(id_group * wp.float32(2 * j /
            EMBED_DIM))
        grid_embeds_prev[i, j] = wp.sin((id_group - wp.float32(1.0)) * wp.
            float32(2 * j / EMBED_DIM))

    w_tile = wp.tile_load(w1, shape=(EMBED_DIM, EMBED_DIM), offset=(0,0))
    y_tile = wp.tile_matmul(grid_embeds_cur, w_tile) # (DIM_IN, EMBED_DIM)
         @ (EMBED_DIM, EMBED_DIM) -> (DIM_IN, EMBED_DIM)
    y_tile_prev = wp.tile_matmul(grid_embeds_prev, w_tile) # (DIM_IN,
        EMBED_DIM) @ (EMBED_DIM, EMBED_DIM) -> (DIM_IN, EMBED_DIM)

    y_tile = wp.tile_map(silu, y_tile) # (DIM_IN, EMBED_DIM)
    y_tile_prev = wp.tile_map(silu, y_tile_prev) # (DIM_IN, EMBED_DIM)
    w2_tile = wp.tile_load(w2, shape=(EMBED_DIM, wp.static(DIM_OUT * K)),
        offset=(0,0))

    y2_tile = wp.tile_matmul(y_tile, w2_tile) # (DIM_IN, EMBED_DIM) @ (
        EMBED_DIM, DIM_OUT * K) -> (DIM_IN, DIM_OUT * K)
    y2_tile_prev = wp.tile_matmul(y_tile_prev, w2_tile) # (DIM_IN,
        EMBED_DIM) @ (EMBED_DIM, DIM_OUT * K) -> (DIM_IN, DIM_OUT * K)

    tvec = wp.vec2(wp.float32(1.0))

    mat20 = wp.vec2(wp.float32(-1.), wp.float32(1.))
    mat21 = wp.vec2(wp.float32(1.0), wp.float32(0.))

    if tid < DIM_IN:
        res = wp.float32(0.0)
```

```
1080        for j in range(DIM_OUT):
1081            id_grid = wp.int32(wp.floordiv(x_tile[i], wp.float32(0.2)))
1082            cvec = wp.vec2(wp.float32(0.0))
1083
1084            t = x_tile[i] / wp.float32(0.2) - wp.float32(id_grid)
1085            start = id_grid % 2
1086            start = wp.where(start < 0, start + 2, start)
1087            elem = int(1)
1088            for cnt in range(start+2, start, -1):
1089                cvec[elem] = wp.where(cnt < wp.static(K), y2_tile_prev[tid, j
1090                    * K + cnt], y2_tile[tid, j * K + cnt])
1091                elem = elem - int(1)
1092
1093            for w in range(0, -1, -1):
1094                tvec[w] = tvec[w+1] * t
1095
1096            res_right = wp.vec2(wp.dot(tvec, mat20),
1097                        wp.dot(tvec, mat21))
1098
1099            res += wp.dot(res_right, cvec)
1100
1101    y[b, tid] = res
```

# E    CLASSIFICATION OVER CIFAR AND IMAGENET

MLP-Mixer is a vision model that replaces both convolutions and self-attention with a simple yet expressive stack of MLPs. An input image is first split into non-overlapping patches, each patch is linearly projected to a fixed-dimensional embedding, and the resulting patch embeddings are arranged as a matrix of shape (number of patches) × (channels). Each Mixer layer then alternates two MLP blocks: a token-mixing MLP that operates across the patch dimension independently for each channel (thereby enabling spatial interaction), and a channel-mixing MLP that operates across channels independently for each patch (enabling feature mixing at each location). Both blocks are implemented with standard fully connected layers, non-linearities, layer normalization, and residual connections, so the entire architecture can be realized using only dense matrix multiplications. Despite its lack of convolutional or attention-based inductive biases beyond patching, MLP-Mixer has been shown to achieve competitive performance with strong CNN and ViT baselines on large-scale image classification benchmarks, making it a natural convolution-free baseline in our experiments.

In this section, we use MLP-Mixer as a convolution-free baseline and replace its token-mixing MLP backbone with KAN and UKAN layers. These experiments mainly demonstrate the large-scale training capabilities of warpKAN on a standard vision architecture by directly swapping in KAN and UKAN layers into existing architectures, and they are not meant to be SOTA models in image classifications. These experiments also act as a design study for future versions of warpKAN, informing our roadmap toward TensorCore–accelerated and tile-based implementations of KAN in warpKAN that further improve efficiency at scale.

We start by training a family of MLP-Mixer baselines on CIFAR-10 and then systematically replacing the token-mixing MLP with KAN or UKAN layers to study their behavior in a standard vision setting. For all models, we fix the backbone and training hyperparameters: the hidden dimension is set to 256, patch size 16x16, and depth 16, the token-mixing expansion factor to 4, channel-mixing expansion factor to 0.5. On top of this fixed backbone, we perform an ablation over KAN/UKAN-specific design choices, in particular the spline grid size and spline order, and measure their impact on training accuracy. This setup isolates the effect of the spline parameters and allows us to directly compare standard MLP-Mixer token mixing with its KAN and UKAN counterparts under matched model capacity. In Table 10, we compare top-1 accuracy of different models.

We observe that in classification tasks KAN slightly outperforms MLP and has better compute/memory fingerprints than UKAN, therefore, on ImageNet-1K (composed of 1.2+M images), we only train MLP and KAN backbone (KAN with order 2 and 3) for 45 epochs. For all models, we choose identical backbone and training hyperparameters as CIFAR-10 model except for the hidden dimension of 512 and patch size of 32x32. The results are shown for Top5 accuracy in Table 11. We

Table 10: Accuracy of different models and configurations on CIFAR-10.

| Model | order | h | Iteration/s | Params (M) | Top-1 Accuracy (%) |
|---|---|---|---|---|---|
| KAN | 3 | 1.0 | 7.86 | 5.5 | 74.0 |
| KAN | 3 | 0.0625 | 6.83 | 1.6 | 72.6 |
| KAN | 2 | 1.0 | 8.71 | 1.5 | 73.7 |
| KAN | 4 | 1.0 | 5.45 | 1.6 | 73.6 |
| MLP | – | – | 26.46 | 1.2 | 73.1 |
| UKAN | 3 | 1.0 | 1.55 | 1.3 | 73.4 |

observe slight improvement in accuracy and motivates development of tiled and TensorCore-based implementations in warpKAN for small grid size. There is also opportunity to fuse CG-MLP into B-Spline component of UKAN, therefore improving its memory and compute fingerprints.

Table 11: Accuracy of different models and configurations on ImageNet-1K.

| Model | order | h | Iteration/s | Params (M) | Top-5 Accuracy (%) |
|---|---|---|---|---|---|
| KAN | 3 | 1.0 | 1.50 | 10.6 | 65.6 |
| KAN | 2 | 1.0 | 1.97 | 10.2 | 66.0 |
| MLP | – | – | 8.48 | 6.6 | 64.9 |

## F  LLM USAGE

We used large-language models (LLMs) as general-purpose assistive tools during the preparation of this paper. Specifically:

- **Writing support:** LLMs were used to improve grammar, clarity, and text flow. All technical content, methodology, experimental design, and conclusions were conceived and verified by the authors.
- **Editing and formatting:** LLMs assisted in rephrasing sentences for readability, generating LaTeX table and figure formatting, and ensuring consistency in notation.
- **Brainstorming:** LLMs were used to explore alternative phrasings, organizational structures, and to verify the completeness of the literature-related sections.

No LLM-generated text or ideas were included without careful review and verification by the authors. The models did not contribute to scientific novelty, research ideation, or experimental results. The authors assume full responsibility for all content in this manuscript.

