# OpenReview forum: "WANT TO TRAIN KANS AT SCALE? NOW UKAN!"
_ICLR.cc/2026/Conference — Submitted to ICLR 2026_

### Official Review · Reviewer_J5gZ · 2025-10-14

**Soundness:** 3
**Presentation:** 3
**Contribution:** 2
**Rating:** 6
**Confidence:** 3

**Summary:**

Introduces UKAN (Unbounded Kolmogorov–Arnold Networks): removes the bounded grid constraint in KANs by generating local B-spline coefficients on-the-fly with a coefficient-generator (CG) MLP conditioned by a positional encoding of grid groups.

Presents warpKAN, a GPU-accelerated implementation that exploits compact B-spline support to cut evaluation from O(K·G·d_in·d_out) to O(K·d_in·d_out) (K = degree+1; G = grid size), reporting 3–30× speedups and up to ~1000× memory reduction vs “vanilla” KANs.

Benchmarks across regression, classification (moons, MNIST), a PINN toy ODE, a toy diffusion model, and molecular property prediction; often matches or beats KAN/MLP baselines.

Notes a clean UKAN→KAN mapping for inference over finite intervals (materialize coefficients).

**Strengths:**

Clear systems contribution: the matrix-form B-spline evaluation and careful GPU design are well-motivated; the roofline analysis is a nice touch.

Practical unboundedness: the CG strategy elegantly avoids brittle grid updates and out-of-support issues while keeping local B-spline stencils.

Broad applicability: results span classic toy problems to ADME prediction; shows UKAN is drop-in across tasks (including swapping into GNN blocks).

Reproducibility extras: explicit complexity table, training details (optimizers/schedules), and appendix with architectural specifics.

**Weaknesses:**

Ablations on UKAN design are thin. How sensitive are results to: CG MLP depth/width, positional-encoding dimension, group size K, grid step h, and the presence/absence of the extra scaling parameter for coefficients? Provide loss-vs-epoch and final accuracy curves for these.

Scope of real-world evals. ADME results are promising but limited to four datasets with fixed featurizers; stronger evidence would include graph featurization end-to-end (e.g., message-passing + UKAN) on larger public sets, with statistical tests and variance reports across multiple splits.

Interpretablity of UKANs? especially it would be good to cite the kan 2.0 paper

**Questions:**

Controlled ablations: (a) CG width/depth × PE dim grid; (b) vary K (spline order) and h; (c) remove the coefficient scale parameter. Report compute/accuracy Pareto.


Harder benchmarks: CIFAR-10/100 with MLP-Mixer-style backbones swapped to UKAN layers; long-sequence regression (e.g., synthetic operators) versus S4/Hyena.


Robustness & OOD: evaluate when inputs shift far outside training support—UKAN should shine here; measure error growth vs distance from training domain.

---

> ### Author Response · Authors · 2025-11-21
>
> We thank the reviewer for their comments. We have included the results of the Biogen ADME dataset as this dataset is one of the cleanest datasets available in this space. However, as these datasets are limited in size, we have also included benchmarks against MoleculeNet datasets in the appendix (Section C). In the appendix, we have used UKAN layers instead of MLPs in an end-to-end manner: (1) node/edge embedding, (2) message-passing, (3) pooling, and (4) prediction modules as outlined in Li, L., Zhang, Y., Wang, G. et al. Kolmogorov–Arnold graph neural networks for molecular property prediction. Nat Mach Intell 7, 1346–1354 (2025). We have also reported results with means, variance, and statistical tests (T-Test for means). MoleculeNet datasets are also much larger than the Biogen ADME datasets. For example, the MUV dataset in Biogen is 30X larger than the largest dataset in Biogen ADME.
> The robustness of the predictions of KANs and UKANs is evaluated as a function of distance to the training set. Both KAN and UKANs show comparable performance characteristics with error increasing with distance to training set. However, UKANs show slightly lower error at larger distances indicating a small improved ability to generalize. All results are discussed in Appendix section C.1.
>
> Incorporating feedback from all reviewers, we also integrated KAN/UKAN into the token-mixing modules of an MLP-Mixer-style architecture for image classification on CIFAR-10 and ImageNet-1K (1.2 M). While this may not be the most optimized usage of KAN/UKAN in vision architectures, the preliminary results are encouraging and demonstrate the feasibility of large-scale training with these models. We report these results in Appendix E. These experiments mainly demonstrate the large-scale training capabilities of warpKAN on a standard vision architecture by directly swapping in KAN and UKAN layers into existing architectures, and they are not meant to be SOTA models in image classifications.

---

> > ### Comment · Reviewer_J5gZ · 2025-11-25
> >
> > Thanks.The author has addressed my concerns, so I have decided to maintain my score.

---

### Official Review · Reviewer_it83 · 2025-10-28

**Soundness:** 2
**Presentation:** 3
**Contribution:** 3
**Rating:** 6
**Confidence:** 2

**Summary:**

This paper tackles two significant limitations of Kolmogorov-Arnold Networks (KANs): their reliance on predefined, bounded grids for B-spline activations, restricting applicability to unbounded domains or requiring data normalization, and the computational inefficiency of existing B-spline evaluation methods, hindering large-scale applications. The authors propose two main contributions: 1) Unbounded KANs (UKANs), which replace the fixed B-spline coefficient grid with a coefficient-generator (CG) MLP that dynamically produces necessary coefficients locally on an unbounded grid, enabling function approximation on unbounded domains without normalization; and 2) warpKAN, a GPU-accelerated library using NVIDIA Warp that implements B-spline evaluation with significantly reduced computational complexity (from $\mathcal{O}(Kd_g d_{in} d_{out})$ to $\mathcal{O}(Kd_{in} d_{out})$) and memory footprint by exploiting local support and basis matrix representations. Benchmarks show warpKAN achieves 3-30x speedups and up to 1000x memory reduction compared to a baseline KAN implementation. UKANs are demonstrated to match or exceed KAN accuracy on regression, classification, physics-informed approximation (PINN), and generative (DDPM) tasks. Finally, the scalability enabled by warpKAN is showcased in a large-scale molecular property prediction task using both accelerated KANs and UKANs.

**Strengths:**

1. The paper directly confronts and provides solutions for two major practical roadblocks for KANs: the bounded grid problem and computational inefficiency, significantly enhancing their usability and scalability.

2. The development of the warpKAN library provides a crucial practical contribution, demonstrating substantial speed and memory improvements through optimized algorithms and GPU acceleration techniques.

3. The effectiveness of UKANs and the scalability benefits of warpKAN are validated across a diverse set of ML tasks (regression, classification, PINN, DDPM, molecular prediction), demonstrating wide applicability.

**Weaknesses:**

1. Introducing the CG-MLP adds complexity and parameters to the UKAN model compared to a standard KAN. The paper lacks a detailed comparison of total parameter counts, training times, and inference latency between KAN and UKAN when achieving similar performance levels on the same task.

2. UKAN introduces new hyperparameters related to the CG-MLP (architecture, embedding dimensions, positional encoding details) and the grid spacing $h$. Guidance on selecting these parameters and sensitivity analysis (especially for $h$) is limited.

3. KANs are often valued for interpretability, but the MLP is lack of interpretability. The introduction of the MLP in KAN is not good for interoretability. For example, how to ensure the accuracy of the coefficients selected by MLP?

**Questions:**

1. Can you elaborate on the potential generalization trade-offs introduced by the CG-MLP in UKANs compared to directly optimizing KAN coefficients?

2. How does the CG-MLP component affect the overall interpretability of the UKAN model compared to standard KANs?
﻿
3. Could you provide guidance or sensitivity analysis regarding the choice of UKAN hyperparameters, particularly the grid spacing $h$ and the CG-MLP design? Could you discuss the computational and memory costs associated with materializing UKAN coefficients for inference, especially in scenarios requiring a large range?

---

> ### Author Response · Authors · 2025-11-21
>
> We appreciate the reviewer’s detailed comments and questions.
>
> (1) Generalization trade-offs and efficiency of CG-MLP vs. direct KAN coefficients
>
> The CG-MLP in UKAN is meant as a generic coefficient generator, analogous to the filter generators used in sequence models such as Hyena, where a simple procedure produces position-dependent convolution kernels. In our case, the CG-MLP replaces a large, static table of B-spline coefficients with a compact function that maps (edge index, knot index) → local spline coefficients.
> From a generalization standpoint, this introduces both a cost and a benefit:
> A standard KAN directly optimizes all spline coefficients as free parameters, which maximizes flexibility but can overfit on small datasets and does not share statistical strength across positions.
> The CG-MLP instead shares parameters across the (potentially unbounded) grid, enforcing a smooth, low-dimensional mapping from indices to coefficients. This acts as an implicit regularizer.
>
> In terms of parameter count and efficiency, when the domain is large, UKAN often reduces the number of learned parameters compared to a KAN coefficient table with the same effective resolution (e.g. the RDKit descriptor model trained with KAN has 2.5B parameters compared to only 6.6M parameters with UKAN). The trade-off is extra compute and activation memory from evaluating the CG-MLP. In the current implementation, this overhead can lead to a  reduced maximum batch size compared to a pure KAN with the same backbone. To mitigate this, we are experimenting with fused UKAN kernels where the CG-MLP evaluation and B-spline basis computation are performed inside a single GPU kernel. A simple prototype of this fused kernel is included in the appendix, and it substantially reduces the overhead of loading generated coefficients.
> In the revised version, we will clarify these trade-offs and include a small comparison (parameter counts, training time, and inference latency) between KAN and UKAN on a representative task.
>
> (2) Effect of CG-MLP on interpretability
> To address the concerns about interpretability, we reproduced Figure 2.4 from the original KAN paper on the regression task . By incorporating entropy loss and magnitude loss on the CG-MLP generated coefficients, we obtain pruned UKAN graphs that closely match the original KAN pruned graphs. As shown in Figure 7 in the appendix of the revised manuscript, the pruned UKAN model clearly preserves the quadratic and sinusoidal behaviors. The resulting structure matches the KAN pruned graph up to a permutation of hidden nodes. We describe the experimental setup and the way we enforce interpretability by decoupling the CG-MLP on a per-edge basis in Appendix A.2.
>
> (3) Hyperparameter guidance (grid spacing h, CG-MLP design) and cost of materializing coefficients
> Grid spacing h.
>
>  We use the same notion of grid spacing as in KAN (distance between adjacent knots) and initialize h following the original KAN settings. Empirically, we find that a slightly larger h than in the original KAN paper often leads to faster and sometimes more stable training for UKAN, because the CG-MLP effectively interpolates over a coarser grid while maintaining good approximation quality.
>
> (4) CG-MLP architecture.
>
>  In all experiments, we intentionally keep the CG-MLP small and fixed across tasks:
> 2 layers with SiLU activations,
> input dimension 2 demb (embedding dimension is either 8 or 16 in all our experiments),
> hidden dimension dembd
> output dimension OutputDim×(order+1), where OutputDim is the number of outgoing channels per edge and “order” is the spline degree.
>
> (5) Computational and memory cost of materializing UKAN coefficients for inference.
>  At inference time, there are two regimes:
>
> On-the-fly UKAN mode: We call the CG-MLP whenever coefficients are needed. This keeps the parameter count small but adds runtime compute and activation memory proportional to the number of queried segments.
>
>
> Materialized KAN mode: For any finite range of interest, we can run the CG-MLP once offline to generate all required B-spline coefficients and store them as a KAN-style coefficient table. In this regime:
>
>
> The computational cost during inference is essentially the same as a standard KAN using warpKAN kernels; the CG-MLP is no longer in the loop.
>
> The memory cost matches that of a KAN with the same grid resolution and range (we trade extra parameters for removing the CG-MLP compute).
> In applications requiring a very large but still finite input range, practitioners can choose between these two modes or a hybrid: use a coarser grid and/or materialize coefficients only over the actual observed data range.
> We hope these clarifications address the reviewer’s concerns about generalization, interpretability, and practical guidance for using UKAN in real applications.

---

> > ### Comment · Reviewer_it83 · 2025-11-24
> >
> > The author has addressed my concerns, so I have decided to maintain my score.

---

### Official Review · Reviewer_wAwE · 2025-10-31

**Soundness:** 3
**Presentation:** 3
**Contribution:** 3
**Rating:** 6
**Confidence:** 4

**Summary:**

This article introduces two main technical innovations regarding KANs. First, they allow KANs to be applied with unbounded grids by using a neural network to encode an infinite number of spline coefficients (only finitely many of which need to be evaluated for any given input), rather than storing the B-spline coefficients themselves as parameters. Second, they provide improved GPU implementations of the spline evaluation which significantly improve computational efficiency. The authors provide numerous numerical experiments demonstrating the effectiveness of their methods.

In my opinion, the paper represents a solid empirical contribution which provides a much more efficient implementation of KANs.

P.S. The title is very creative!

**Strengths:**

Training KANs is an important problem. This paper makes substantial progress in developing more efficient and flexible implementations of KANs.

The numerical results look very strong. The runtimes remain nearly constant even for very large grids in Table 2, which demonstrates the parallelism of their method. The UKAN method consistently outperforms vanilla KANs for wide variety of problems, including practical drug discovery problems.

**Weaknesses:**

The paper doesn't contain any theoretical analysis, although this is to be expected given the empirical nature of the contribution.

I also hope the authors will release their code publicly if the paper is accepted.

**Questions:**

I don't quite understand equation (8). Is u supposed to be a position in $\mathbb{R}$ or a vector? It seems from the definition following the equation that u has components for each i. Is this correct? Does this mean that u is a vector? Please provide a bit more careful explanation of equation (8).

---

> ### Author Response · Authors · 2025-11-21
>
> We thank the reviewer for their comments.
>
> Regarding Eq. (8) and the definition of u. In Eq. (8), u denotes a scalar input coordinate for a 1D B-spline basis function. In the network, this scalar is defined per edge: for each connection from neuron iin layer l to neuron j in layer l+1, we evaluate a B-spline spline(u_(i,j)^((l) ) )at a scalar argument u_(i,j)^((l))(obtained from the activation at node iin layer l). We have revised the manuscript to make this explicit by writing u_(i,j)^((l))and clarifying that Eq. (8) is the general 1D spline definition, which we then instantiate per edge in the network. We also cleaned up the surrounding text to avoid any suggestion that uis a vector.
>
> On code release and broader impact: We will release our implementation under an Apache-2.0 license and agree that this code and its future extensions will enable larger-scale KAN/UKAN training. Motivated by our large-scale image experiments, we are currently exploring tiled KAN/UKAN variants and TensorCore implementations. We will include a link to the public repository in the final version (consistent with anonymity requirements).

---

### Official Review · Reviewer_K2Jk · 2025-11-01

**Soundness:** 3
**Presentation:** 1
**Contribution:** 2
**Rating:** 4
**Confidence:** 4

**Summary:**

This paper addresses two bottlenecks when using Kolmogorov-Arnold Networks (KANs): their computational inefficiency and their reliance on predefined, bounded grids.

The authors present two core contributions:
- A high-performance, GPU-accelerated library (using NVIDIA Warp) that re-implements B-spline evaluation.
- A new architecture that removes the need for a bounded grid. It uses a "coefficient-generator" (CG) model to dynamically generates the required B-spline coefficients on the fly.

**Strengths:**

The paper's strength lies in its both engineering and architecture contributions. Both the warpKAN library and the UKAN architecture  extends KAN's capabilities.

The performance benchmarks for warpKAN are not just marginal improvements; a 3-30x speedup and 1000x memory reduction represent a step-change in feasibility.

**Weaknesses:**

UKAN Re-introduces a "Black Box": A primary appeal of KANs over MLPs is their potential for greater interpretability (i.e., "white-box" view of learnable spline coefficients). The UKAN architecture, by using an MLP (the CG) to generate these coefficients, re-introduces a black-box component. The claim that interpretability is retained by "materializing" coefficients post-training feels like a workaround, as the reason for those coefficient values is now hidden inside the CG-MLP.

Overhead of the Coefficient-Generator (CG): The paper introduces the computational cost of the CG-MLP (O_CG in Table 1) but does not deeply analyze the practical trade-offs. For a given problem, it's unclear when the computational overhead of running this new MLP (at every forward pass for every input) becomes more expensive than simply using a larger, fixed grid enabled by warpKAN.

Not really scalable experiments: The experiments remain relatively small and rely on the same dataset used in the original KAN paper—a choice that has been repeatedly criticized by the community. To substantiate the claim of “TRAIN KANs AT SCALE”, it is essential to evaluate performance on large-scale, real-world datasets in domains such as image, audio, or language.

**Questions:**

See above

---

> ### Author Response · Authors · 2025-11-21
>
> We thank the reviewer for their helpful comments and have revised the manuscript accordingly.
>
> Interpretability:
> To address the concerns about interpretability, we reproduced Figure 2.4 from the original KAN paper on the regression task . By incorporating entropy loss and magnitude loss on the CG-MLP generated coefficients, we obtain pruned UKAN graphs that closely match the original KAN pruned graphs. As shown in Figure 7 in the appendix of the revised manuscript, the pruned UKAN model clearly preserves the quadratic and sinusoidal behaviors. The resulting structure matches the KAN pruned graph up to a permutation of hidden nodes. We describe the experimental setup and the way we enforce interpretability by decoupling the CG-MLP on a per-edge basis in Appendix A.2.
>
> Real-world applicability and scale: We further demonstrate the practical relevance of KAN/UKAN by evaluating them on Biogen ADME datasets comprising four ADME (Absorption, Distribution, Metabolism, and Excretion) endpoints. These are central to small-molecule drug discovery pipelines, where accurate in silico prediction can substantially accelerate candidate prioritization. This experiment also highlights the scalability of our approach: the KAN model used in this setting has 2.56B parameters, whereas the corresponding UKAN model has only 6.66M parameters. Additional details on model scale are provided in Appendix A.6.
>
>
> Large-scale training with vision backbones: Incorporating feedback from all reviewers, we also integrated KAN/UKAN into the token-mixing modules of an MLP-Mixer-style architecture for image classification on CIFAR-10 and ImageNet-1K (1.2 M). While this may not be the most optimized usage of KAN/UKAN in vision architectures, the preliminary results are encouraging and demonstrate the feasibility of large-scale training with these models. We report these results in Appendix E. These experiments mainly demonstrate the large-scale training capabilities of warpKAN on a standard vision architecture by directly swapping in KAN and UKAN layers into existing architectures, and they are not meant to be SOTA models in image classifications.
>
>
> We also added a section for comparison of KAN and UKAN both from warpKAN, where we observed UKAN consume more memory and compute compared to our optimized KAN implementations, we also provided prototype code in warp to address both problems for UKAN in Appendix D with fusing CG-MLP into B-spline evaluation kernel, so both problems are fixable given a fixed CG-MLP architecture. This implies for small batch sizes and wide network architecture, it might be beneficial to switch to UKAN and for larger batch sizes switch to KAN.

---

### Meta-Review · Area_Chair_Bog2 · 2026-01-06

**Summary:**

All the reviewers acknowledge the importance of the problem the authors address in the paper. However, the proposed approach raises the following concerns.
- The lack of large-scale experiments despite the paper claiming to allow for training KANs at a large scale (Reviewers K2Jk, J5gZ).
- Predicting the spline coefficients with an MLP removes the interpretability of KANs --- one of their attractive properties (Reviewers K2Jk, it83, J5gZ).
- Insufficient ablation study and comparisons in terms of the computational budget (Reviewers K2Jk, it83, J5gZ).
- Lack of theoretical analysis and the open-source implementation (Reviewer wAwE).

**Reviewer Concerns:**

During the rebuttal, the authors updated the manuscript, adding additional empirical evaluation to the appendix.
- For large-scale experiments, they added an empirical study on training a classification model on CIFAR-10 and ImageNet datasets. For the baseline, the authors replace the MLP-Mixer's layers with KAN and UKAN architectures. However, the performance of these models is questionable. For instance, the classification accuracy of the baseline model (MLP-mixer) on CIFAR-10 is $73.1\%$, which is far from being relevant.
- For the interpretability, the authors visualize the final architecture with the predicted coefficients in Figure 7. However, this does not address the concerns of reviewer K2Jk.
- Finally, the requested ablation study is not provided. The authors qualitatively compare the models in their response using vague phrasing like "small, large, compact".

**Reviewer Scores:**

I do not see the reason for a significant increase in scores after the rebuttal. I do not find the authors' rebuttal convincing, and I assume the reviewers could only maintain their scores rather than reduce them.

---

### Decision · Program_Chairs · 2026-01-26

Reject